# Impact of Marker Pruning Strategies Based on Different Measurements of Marker Distance on Genomic Prediction in Dairy Cattle

**DOI:** 10.3390/ani11071992

**Published:** 2021-07-02

**Authors:** Duanyang Ren, Jinyan Teng, Shuqi Diao, Qing Lin, Jiaqi Li, Zhe Zhang

**Affiliations:** Guangdong Laboratory of Lingnan Modern Agriculture/Guangdong Provincial Key Lab of Agro-Animal Genomics and Molecular Breeding, College of Animal Science, South China Agricultural University, Guangzhou 510642, China; ren457842071@163.com (D.R.); kingyan312@live.cn (J.T.); saradiao@126.com (S.D.); qing_lin1996@126.com (Q.L.); jqli@scau.edu.cn (J.L.)

**Keywords:** genomic prediction, marker density, genetic distance, physical distance, Holstein dairy cattle, high-density SNP data

## Abstract

**Simple Summary:**

The usefulness of genomic prediction (GP) has been widely proofed by breeding analysis in livestock, plants and aquatic populations. It is well known that ‘marker density’ is a critical factor that affects the accuracy of GP, however, how to properly measure ‘marker density’ in GP is yet to be determined. With population-level whole-genome sequence data or high-density single nucleotide polymorphism (SNP) data available, this question seems to be answered more convincingly. In this study, we investigated and discussed the impact of four ‘marker density’ measures that reflect genetic or physical distances between SNPs on the accuracy of GP in a Germany Holstein dairy cattle population. Our results showed that the degree of variation of physical distance between adjacent SNPs had significant effects on the accuracy of GP, while the genetic distance between SNPs had no relationship with the accuracy of GP. Therefore, for studies based on high-density SNP data, the default strategy of pruning SNPs based on genetic distance is detrimental to heritability estimation and genomic prediction. The results extended the communities knowledge of ‘marker density’ and provided useful suggestions for the application and research on genome prediction.

**Abstract:**

With the availability of high-density single-nucleotide polymorphism (SNP) data and the development of genotype imputation methods, high-density panel-based genomic prediction (GP) has become possible in livestock breeding. It is generally considered that the genomic estimated breeding value (GEBV) accuracy increases with the marker density, while studies have shown that the GEBV accuracy does not increase or even decrease when high-density panels were used. Therefore, in addition to the SNP number, other measurements of ‘marker density’ seem to have impacts on the GEBV accuracy, and exploring the relationship between the GEBV accuracy and the measurements of ‘marker density’ based on high-density SNP or whole-genome sequence data is important for the field of GP. In this study, we constructed different SNP panels with certain SNP numbers (e.g., 1 k) by using the physical distance (PhyD), genetic distance (GenD) and random distance (RanD) between SNPs respectively based on the high-density SNP data of a Germany Holstein dairy cattle population. Therefore, there are three different panels at a certain SNP number level. These panels were used to construct GP models to predict fat percentage, milk yield and somatic cell score. Meanwhile, the mean ($\bar{d}$) and variance ($\sigma_{d}^{2}$) of the physical distance between SNPs and the mean ($\bar{r^{2}}$) and variance ($\sigma_{r^{2}}^{2}$) of the genetic distance between SNPs in each panel were used as marker density-related measurements and their influence on the GEBV accuracy was investigated. At the same SNP number level, the $\bar{d}$ of all panels is basically the same, but the $\sigma_{d}^{2}$, $\bar{r^{2}}$ and $\sigma_{r^{2}}^{2}$ are different. Therefore, we only investigated the effects of $\sigma_{d}^{2}$, $\bar{r^{2}}$ and $\sigma_{r^{2}}^{2}$ on the GEBV accuracy. The results showed that at a certain SNP number level, the GEBV accuracy was negatively correlated with $\sigma_{d}^{2}$, but not with $\bar{r^{2}}$ and $\sigma_{r^{2}}^{2}$. Compared with GenD and RanD, the $\sigma_{d}^{2}$ of panels constructed by PhyD is smaller. The low and moderate-density panels (< 50 k) constructed by RanD or GenD have large $\sigma_{d}^{2}$, which is not conducive to genomic prediction. The GEBV accuracy of the low and moderate-density panels constructed by PhyD is 3.8~34.8% higher than that of the low and moderate-density panels constructed by RanD and GenD. Panels with 20–30 k SNPs constructed by PhyD can achieve the same or slightly higher GEBV accuracy than that of high-density SNP panels for all three traits. In summary, the smaller the variation degree of physical distance between adjacent SNPs, the higher the GEBV accuracy. The low and moderate-density panels construct by physical distance are beneficial to genomic prediction, while pruning high-density SNP data based on genetic distance is detrimental to genomic prediction. The results provide suggestions for the development of SNP panels and the research of genome prediction based on whole-genome sequence data.

## 1. Introduction

Implementing genomic selection can increase genetic gain, which has now been demonstrated in livestock [1,2], plants [3,4] and aquatic [5,6]. Next-generation sequencing efforts have uncovered the genome sequences of many species and revealed thousands of single-nucleotide polymorphism (SNP) markers, making the genomic prediction (GP) based on high-density SNP or whole-genome sequence (WGS) data possible [7,8]. High-density SNP panels will be the future’s genotype data in genomic selection [2,8]. Therefore, it is necessary to explore factors affecting the performance of GP based on high-density SNP data. In the genetic analysis based on high-density panels, markers were usually thinned by linkage disequilibrium (LD) threshold (genetic distance) [9,10], while moderate or low-density panels are generally constructed by SNPs distributed evenly in the whole genome [11,12]. However, the effect of marker selection methods (physical or genetic distance) on the genomic prediction performance needs to be studied in detail.

Compared with low-density (<10 k SNPs) and moderate-density (10–100 k SNPs) panels, high-density panels (>100 k SNPs) can improve the accuracy of GP [8,13]. However high-density SNP genotyping of all selected candidates in each generation may not be cost-effective. Habier et al. [11] proposed to genotype key individuals or ancestors with a high-density panel and candidates or dams with an evenly spaced, low-density panel. The low-density panels are then imputed to high-density panels. The accuracy of genomic prediction is greatly improved by pooling animals with real and imputed high-density SNP data [14]. This strategy is currently the primary method for genomic prediction using high-density SNP data. With the development of imputation methods applicable to different species and population structures [15,16,17], and the availability of resources that allow imputing SNP arrays to high-density data or WGS, such as the 1000 Bull Genome Project [7], low-density panels can be accurately imputed to high-density panels. At the same time, by considering the influence of SNP number, physical distance and genetic distance between SNPs on the imputation accuracy, some cost-effective low-density panels have been designed [18,19]. In general, the accuracy of imputed genotypes is about 90% [20,21], and the genomic estimated breeding value (GEBV) accuracies are very similar for imputed and real high-density panels [1,22]. Therefore, the imputed high-density panels provide more possibilities for genomic prediction research, such as the application in the dairy cattle population [23].

Recently, several studies have found that compared with moderate-density panels, GEBV accuracy based on real or imputed high-density panels has decreased [24,25]. The decrease may result from genotype imputation errors [26], the limited population size [27], or the noise involved in high-density data [28,29]. The decrease in accuracy could also indicate that, for some traits, the high-density panel is not necessarily optimal in GP. To make the information of the high-density SNP data more effective, it is necessary to clarify the relationship between measurements of ‘marker density’ and the GEBV accuracy. Some studies have found that the increase in GEBV accuracy is very limited after a certain number of SNPs are reached [5,8,30]. Therefore, in addition to the SNP number, some measurements of ‘marker density’, such as the physical distance and genetic distance between adjacent SNPs, may also affect the performance of GP. Although the effect of these measurements of marker density on genotype imputation accuracy has been validated [31], the effect on GEBV accuracy remains unclear.

It is known that GEBV accuracy is closely related to marker density, however, it is yet to be determined the key measurements of ‘marker density’. Therefore, we conducted this study with the aims of (1) evaluating the impact of ‘marker density’ measured by physical or genetic distance on the performance of genomic prediction in a dairy cattle population, and (2) exploring the effect of mean and variance of marker distances on the accuracy of GP. Our result extended knowledge of measurements of ‘marker density’ on the performance of GP.

## 2. Materials and Methods

### 2.1. Dataset

The German dairy cattle population including 2000 bulls from Vereinigte Informationssysteme Tierhaltung Wirtschaftlicher Verein was used in this study [32], and all individuals were genotyped with the Illumina Bovine SNP50 Beadchip [33]. The 50 k SNP data were imputed to 770 k via Beagle software [16] with the high-density SNP data obtained from the 1000 Bull Genome Project [7] as reference. SNPs with either minor allele frequency < 0.005, or genotype call rate < 0.9, or *p-*value < 0.01 from the Hardy–Weinberg equilibrium test were removed, after filtering, 335,753 SNPs were retained for further analyses (Appendix A). The distribution of the physical distance and genetic distance between adjacent SNPs are shown in Appendix A. Highly reliable traditional estimated breeding values (EBVs) of three traits, milk fat percentage (FP), milk yield (MY), and somatic cell score (SCS), were available for all bulls. The three traits may represent different types of genetic architectures that are composed of (1) one or few major genes and many loci of small effect (FP), (2) some loci of moderate effect and many loci of small effect (MY), and (3) many loci of small effect (SCS). EBVs of the three traits were used as phenotypes in the GP model.

### 2.2. Measurements of Marker Distance

In this study, three methods, namely physical distance (PhyD), genetic distance (GenD) and random distance (RanD), were used to measure the marker distance. The PhyD method selects markers by limiting the number of base pairs (bp) between adjacent SNPs. The GenD method filters the SNPs by limiting the maximum *r*^2^ (the multiple correlation coefficient for an SNP being regressed on all other SNPs simultaneously) between SNPs. The RanD method was implemented using PLINK software. The detailed implementation of the methods is shown in the Appendix B.

In the random method, 0.3%, 0.6%, 0.9%, 1.2%, 1.5%, 2.1%, 3%, 4.5%, 6%, 9%, 15%, 22.5%, 30%, 60% of the 335,753 SNPs were randomly sampled, and panels with 1, 2, 3, 4, 5, 7, 10, 15, 20, 30, 50, 75, 100, and 200 k SNPs were constructed. We term the marker sets containing 1–10 k SNPs as the low-density panels, containing 15–75 k SNPs as moderate-density panels and containing 100–300 k SNPs as the high-density panels. For the PhyD method, we plotted the relation curve between the number of the selected SNPs and the threshold of the distance (bp) between adjacent SNPs in Figure 1A. The threshold of distance limits the minimum physical distance between adjacent SNPs. The GenD method was implemented in the PLINK software [34], and the SNP pruning process is: (a) calculate *r*^2^ between each pair of SNPs in the window, (b) remove one of a pair of SNPs if the *r*^2^ is greater than the threshold, (c) shift the window a step length forward and repeat the procedure. The window size and step were set as 50 and 5 kb respectively, and the relation curve between the number of selected SNPs and the threshold of genetic distance (*r*^2^) is present in Figure 1B. In the PhyD and GenD methods, appropriate thresholds of physical distance and genetic distance were selected to construct panels with SNP number consistent with the random method (Appendix A). For the GenD method, only 75 k SNPs were selected even the *r*^2^ is close to 1. As a result, the GenD method does not produce panels containing 100 or 200 k SNPs. According to the above three methods, a total of 40 panels were generated, which can be regarded as SNP chips with different SNP numbers and conduct according to different marker selection methods.

### 2.3. Genomic BLUP Model

All genomic predictions in this study were based on the genomic BLUP (GBLUP) model, the model includes a single random genetic effect and was expressed as:(1)y=μ+Zg+e
where y is the vector of EBVs, μ is a vector of overall mean, g is a vector of individual genetic values captured by all SNPs in the panel, Z is the design matrix of genetic values, and e is a vector of residuals. The random genetic and residual values are assumed to be independent normally distributed values described as: $g\sim N(0,G\sigma_{g}^{2})$ and $e\sim N(0,I\sigma_{e}^{2})$, $\sigma_{g}^{2}$ and $\sigma_{e}^{2}$ are the additive genetic variance and residual variance. In this study, all variance components were estimated using the average information restricted maximum likelihood (AI-REML) algorithm in BLUPF90 software [35].

The additive **G** matrix was constructed using all SNPs in the panel [36]:(2)$$G=\frac{WW^{\prime }}{2\sum_{k=1}^{K}p_{k}(1–p_{k})}$$
where W=M–2P is the centered genotype matrix, M is the genotype matrix with elements of 0, 1 and 2, P is a matrix of $1\times p_{k}$ in the *k*th column, and $p_{k}$ is the frequency of a given allele of the *k*th SNP. The solution of $\hat{g}$ is equal to ${\left(Z^{\prime }R^{-1}Z+G^{-1}\right)}^{-1}Z^{\prime }R^{-1}(y–\hat{\mu })$, where $R=I\sigma_{e}^{2}$ is the covariance matrix of the residuals. Each of the 40 panels produced in the Marker selection methods section was used to construct the **G** matrix separately.

### 2.4. Model Assessment

The predictive performance of the GBLUP model was evaluated using the 10-fold cross-validation (CV) method. A 10-fold CV process is as follows: (1) Randomly split all bulls into 10 folds. (2) Build the GP model on nine folds of the bulls (training set), and the remaining fold (validation set) is used to test the accuracy of the prediction model. (3) Repeat (2) until each of the 10-folds has served as the validation set. (4) The Pearson’s correlation coefficient of EBV and GEBV for individuals in the 10 validation sets obtained above is called the CV accuracy and will serve as the prediction accuracy of the GBLUP model. This 10-fold CV process was repeated 10 times for each scenario. Therefore, the 10 × 10 fold CV was used to evaluate GEBV accuracy in our study. The variance component estimates calculated by the above CV process were also used for subsequent analysis.

### 2.5. Measurements of Marker Density

To make it easier to compare the panels produced by different marker selection methods, four statistics were calculated for each SNP panel, i.e., the mean ($\bar{d}$) and variance ($\sigma_{d}^{2}$) of the physical distance between adjacent SNPs and the mean ($\bar{r^{2}}$) and variance ($\sigma_{r^{2}}^{2}$) of the *r*^2^ (genetic distance) between adjacent SNPs. These four statistics were used as the measurements of marker density. For each of the 40 panels, the four statistics need to be calculated separately. In this study, the length of the chromosomes used to construct the various panels was fixed. At the same SNP number level, the $\bar{d}$ of all panels is basically the same, but $\sigma_{d}^{2}$, $\bar{r^{2}}$ and $\sigma_{r^{2}}^{2}$ are different. Therefore, we only investigated the effects of $\sigma_{d}^{2}$, $\bar{r^{2}}$ and $\sigma_{r^{2}}^{2}$ on the GEBV accuracy.

## 3. Results

### 3.1. Measurements of Marker Density for Different Panels

A total of 40 panels were generated by PhyD, GenD and RanD methods. The distributions of the physical distance and genetic distance between adjacent SNPs for each panel are shown in Figure 2. The physical distance between adjacent SNPs was replaced by logarithm base 10 because the distance is of different orders of magnitude at different SNP number levels. Compared with the other two methods, the panel constructed by PhyD has the smallest $\sigma_{d}^{2}$. For the panels constructed by the GenD or RanD method, the $\sigma_{d}^{2}$ was large, this is most obvious in the panels with few SNPs constructed by the GenD method (Figure 2A). The $\bar{r^{2}}$ and $\sigma_{r^{2}}^{2}$ increase with the number of SNPs in the panels, the RanD method increases faster, followed by the PhyD method, and the GenD method is the slowest (Figure 2B).

### 3.2. Accuracies of Genomic Prediction

The genomic prediction performance of panels constructed by different methods is shown in Figure 3. Compared with GenD and RanD, the low-density panels constructed by PhyD method have advantages in genomic prediction for all three traits (Figure 3). The GEBV accuracy of moderate and low-density panels constructed by PhyD method was 3.8~34.8% higher than that of moderate and low-density panels constructed by RanD or GenD method. The panel with the highest SNP number did not achieve the best performance of genomic prediction for all traits. For SCS controlled by many loci of small effect, the panel with 30 k SNPs constructed by GenD method corresponds to the highest GEBV accuracy, which is 0.8% higher than that of the highest density SNP panel, but there is no obvious difference in GEBV accuracy between the 30 k panel constructed by PhyD (Figure 3A). For MY and FP affected by loci of moderate or large effect [37], the GEBV accuracy based on panels constructed by PhyD method with about 20 k SNPs was the highest, which was 1.5% and 8% higher than that of the highest density panel respectively (Figure 3B,C). The GEBV accuracy of moderate density panels (20–30 k SNPs) constructed by PhyD was the same as or slightly higher than that of high-density SNP panels.

### 3.3. Variance Component Estimates

The genetic variance estimates based on different panels correspond to the GEBV accuracy, that is, the higher the estimate of genetic variance, the higher the GEBV accuracy. In other words, the more accurate the GP model, the more genetic variance it can capture, and the lower the estimate of residual variance (Figure 4). Therefore, the estimates of the residual variance based on different panels are contrary to the trend of GEBV accuracy. The genetic variance estimates based on low-density panels constructed by PhyD method are 5~51.6% higher than that of low-density panels constructed by RanD or GenD method.

### 3.4. Relationships between Measurements of Marker Density and GEBV Accuracy

Panels at the same SNP number level constructed by different methods have different performances in genomic prediction, and this difference is most obvious in the low and moderate-density panels. Therefore, we investigated the relationship between marker density-related measurements and GEBV accuracy in low and moderate-density panels. Table 1 lists the marker density-related measures and GEBV accuracy for low and moderate-density panels constructed by PhyD, GenD or RanD. Compared with GenD and RanD, the panels constructed by PhyD have the minimum $\sigma_{d}^{2}$ and generally have higher GEBV accuracy than the panels constructed by GenD and RanD at each SNP number level. Panels constructed by GenD had much smaller $\bar{r^{2}}$ or $\sigma_{r^{2}}^{2}$ at each SNP number level than those constructed by RanD, but had no obvious advantage or disadvantage in terms of GEBV accuracy compared to those constructed by RanD. With the increase of the SNP number in the panels, the $\bar{r^{2}}$ and $\sigma_{r^{2}}^{2}$ of panels constructed by PhyD changed greatly compared with those constructed by GenD or RanD, but the GEBV accuracy of the panel constructed by PhyD was generally superior to that of the panels constructed by GenD and RanD.

To further investigate the relationship between marker density-related measurements and GEBV accuracy, we tested the correlation between each measurement and GEBV accuracy. First, the three values of each marker density-related measure at each SNP number level were standardized as follows:(3)$$z_{i}=\frac{x_{i}-\mathrm{mean}(x)}{\mathrm{sd}(x)}.$$
where $x_{i}$ is the *i*th variable of x (refers to any of the marker density-related measurements), mean(x) is the mean value of x, sd(x) is the standard deviation of x, $z_{i}$ is standardized $x_{i}$. The three GEBV accuracies for each trait at each SNP number level were also standardized by Equation (3). The measurements and GEBV accuracies were standardized in each SNP number level respectively. Then the correlation analysis was carried out by combining the information of all SNP number levels in Table 1. Each standardized measurement value (standardized values in any column $\sigma_{d}^{2}$, $\bar{r^{2}}$ and $\sigma_{r^{2}}^{2}$ in Table 1) corresponds to three standardized GEBV accuracies (corresponding GEBV accuracies in columns FP, MY and SCS in Table 1). Therefore, a total of 99 pairs of measurement values and GEBV accuracies were used to detect the relationship between each marker density-related measure and GEBV accuracy. The correlation coefficient between the marker density-related measurements and the GEBV accuracy was calculated by Pearson’s correlation. The *t*-value of the correlation coefficient was used as the test statistic for the significance test. In this study, the correlation analysis was used to detect the relationship between marker density-related measures and GEBV accuracy under a certain SNP number (but combining information from different SNP number levels), rather than detect the influence of marker density-related measures on GEBV accuracy as the SNP number changes, so we first standardized the measurements and the GEBV accuracies in each SNP number level. Another purpose of our standardization (using standard deviation) is to make the information at each SNP number level contribute equally to the results of the correlation analysis.

Correlation test results showed that there was a significant negative correlation between $\sigma_{d}^{2}$ and GEBV accuracy, and the correlation coefficient between them was −0.83 (*p* < 0.001) (Figure 5A). Compared with GenD and RanD, panels constructed by PhyD have a very small $\sigma_{d}^{2}$ (Table 1), and the standardized $\sigma_{d}^{2}$ are mainly clustered at −1 (Figure 5A). Panels constructed by GenD and RanD have larger $\sigma_{d}^{2}$ (Table 1), with standardized $\sigma_{d}^{2}$ clustering around 0.5 (Figure 5A). In Figure 5B,C, the standardized $\bar{r^{2}}$ and $\sigma_{r^{2}}^{2}$ near −1 were generated by GenD, and the standardized $\bar{r^{2}}$ and $\sigma_{r^{2}}^{2}$ near 1 were generated by RanD. Although the $\bar{r^{2}}$ and $\sigma_{r^{2}}^{2}$ of panels constructed by GenD and RanD are significantly different, they do not affect the GEBV accuracy. In Figure 5B,C, the standardized $\bar{r^{2}}$ and $\sigma_{r^{2}}^{2}$ between −0.5 and 0.5 were generated by PhyD. Compared with GenD and RanD, panels constructed by PhyD have higher GEBV accuracy in In Figure 5B,C, which seems to be caused by other factors, namely $\sigma_{d}^{2}$, independent of $\bar{r^{2}}$ and $\sigma_{r^{2}}^{2}$. For trait SCS, there seems to be a consistent negative correlation between the GEBV accuracy and $\bar{r^{2}}$ (or $\sigma_{r^{2}}^{2}$) for panels at 15 to 50 k (Table 1). We combined the information of 15 to 50 k SNP panels to test the correlation between GEBV accuracy of SCS and the marker density-related measurements, the results are shown in Appendix A. In moderate-density panels, $\sigma_{d}^{2}$ has a weak negative correlation with GEBV accuracy of SCS (the correlation coefficient was −0.54, *p* = 0.073, Appendix A); $\bar{r^{2}}$ and $\sigma_{r^{2}}^{2}$ were negatively correlated with GEBV accuracy of SCS (correlation coefficient were close to −0.8, *p* < 0.01, Appendix A). This may be related to the higher GEBV accuracy of the moderate-density panels constructed by GenD in SCS (Figure 3A).

The above results seem to indicate that the variance of physical distance between adjacent SNPs in panels influences the GEBV accuracy, and we further verified this result. We generated four different marker sets regardless of the physical distance and genetic distance between the SNPs. The first marker set (Scenario 1) was constructed as follow: (1) The SNPs were numbered 1 to 335,753 in the order in which they are placed on the chromosome, and select the SNP numbered 1, (2) select the SNP numbered *i* + 67, and *i* is the SNP number selected last time, (3) repeat (2) until throughout the entire genome. For the construction of marker sets 2–4 (Scenario 2–4), just replace the 67 in step 2 with integer sampling from normal distributions with a mean of 67 and variance of 20, 40 or 80 respectively. The $\bar{d}$ and $\sigma_{d}^{2}$ of scenario1–4 are listed in Appendix A. It can be seen that the four marker sets have the same $\bar{d}$ and different $\sigma_{d}^{2}$. Therefore, these four scenarios were used to study the relationship between GEBV accuracy and $\sigma_{d}^{2}$. The performance of the marker sets of scenarios 1–4 on genomic prediction was verified with the 10 × 10 fold CV. Figure 6 shows that the GEBV accuracy decreases with the increase of $\sigma_{d}^{2}$.

## 4. Discussion

Marker density is an important factor affecting the results of genomic prediction and heritability estimation. In general, the low-density SNP chips for genome prediction were constructed by SNPs evenly distributed across the entire genome [12]. High-density SNP or genome-wide sequence data used for genome-wide association analysis (GWAS) and heritability estimation are usually thinned by default using LD thresholds between markers [9,10]. Previous studies seem to have ignored the effects of different measurements of marker density, namely physical distance and genetic distance and their characteristics, on the results of genome analysis. In this study, the influences of measurements of ‘marker density’ were investigated systematically using high-density SNP data from a Holstein cattle population. We found that the degree of variation of physical distance between SNPs had significant effects on the accuracy of GP and the estimation of genetic variance. However, the genetic distance between SNPs was not significantly correlated with the accuracy of genomic prediction.

### 4.1. Relationship between Measurement of “Marker Density” and GP Performance

Previous studies have found that the marker selection methods have an impact on the distribution of SNPs on the chromosome and, in turn, the performance of GP [19]. In this study, the distance between adjacent SNPs was measured by physical or genetic map. For panels with a certain SNP number, $\sigma_{d}^{2}$ is an important factor found in this study that significantly affects the GEBV accuracy. $\sigma_{d}^{2}$ is the measure of physical distance variation between adjacent SNPs. The $\sigma_{d}^{2}$ of the panels constructed by the PhyD method is far less than that of the panels generated by RanD or GenD methods (Table 1). The genomic prediction performance based on panels constructed by the PhyD method is almost always better than that of the panels constructed by the other two methods (Figure 3). Correlation test between GEBV accuracy and $\sigma_{d}^{2}$ shows that there is a negative correlation between them (Figure 5A), and this relationship was verified in four arbitrary panels that with different $\sigma_{d}^{2}$ (Appendix A, Figure 6). Werner et al. [4] found that the GEBV accuracy based on the low-density panels constructed by GWAS results was lower than that of the panels randomly constructed in some traits. This may be due to uneven distribution of the GWAS peak on the chromosome and resulting in larger $\sigma_{d}^{2}$. Meanwhile, The low-density panels constructed by minor allele frequency (MAF) filtering were worse than randomly constructed panels in GP (Figure 2 in their paper) may also related to larger $\sigma_{d}^{2}$ [38]. The measurements of genetic distance between adjacent SNPs are not related to the GEBV accuracy for low and moderate-density panels (Figure 5B,C), although the genomic prediction is dependent on the LD between SNP and quantitative trait locus [39,40]. Because the physical distance between adjacent SNPs of panels generated by the GenD method varies greatly (Table 1), the GenD method is not more favorable or worse than the random method when constructing low-density panels (Figure 3). The previous study has also found that low-density panels constructed by genetic distance have large gaps [31]. In the studies of GWAS and SNP-heritability estimation based on high-density SNP data or whole-genome sequence data, *r*^2^ is usually used as a threshold to prune the SNPs before formal analysis [9,10]. However, even a threshold of 0.98 would produce moderate-density panel (Appendix A) with large $\sigma_{d}^{2}$ (Table 1), resulting in a loss of genetic variance (Figure 4).

### 4.2. Development of Cost-Effective Panels

The genomic selection currently relies on collecting genome-wide genotype data across a large number of individuals, which requires substantial economic investment. For small and medium-sized enterprises in livestock and poultry breeding, a cost-effective low-density panel is needed [5]. The GEBV accuracy increases rapidly with the number of SNPs in the low-density panel, however, as the number of SNPs increases to moderate or high density, the improvement of GEBV accuracy is very limited (Figure 3). Previous studies have also reported consistent results [5,30]. Therefore, it seems to be feasible and cost-effective to develop a low-density panel according to the above characteristics. The low-density panel constructed by the PhyD method can minimize the variance of the distance between adjacent markers and allow GEBV accuracy to be maintained as much as possible with a limited SNP number. Meanwhile, the equidistant distribution of SNPs is also conducive to genotype imputation [12]. Compared to moderate and high-density panels (more than 20 k SNPs), the decrease in GEBV accuracy and genetic variance estimates based on low-density panels (less than 10 k SNPs) constructed by the PhyD method seems to be independent of trait genetic architecture (Figure 3 and Figure 4), which has also been reported in the previous studies [11,41]. Therefore, the PhyD method can be used as an effective method to construct moderate and low-density panels. We found that when the SNP number reached moderate density, the increase of GEBV accuracy is very limited for all dairy cattle traits (Figure 3). The definition of low-density, moderate-density and high-density panels may vary with species and the number of high-density markers available, here we illustrate our recommendations based on the dairy cattle data. It may be an effective strategy to develop the 10 k low-density panel by the PhyD method for dairy cattle, which can maintain the GEBV accuracy. The cost savings of the 10 k panel compared to the commonly used 50 k panels will facilitate the application of GP. In fact, most panels used for GP were designed with markers evenly distributed across the entire genome [11,12]. At the same time, it is a good strategy to add a small amount of trait-related causative variations to the low-density panel [4].

### 4.3. Pruning Strategies of High-Density SNP Data in Genome prediction

Among all traits in this study, it is not the panel with the highest density that can capture the most amount of genetic variance (Figure 4) and obtain the highest GEBV accuracy (Figure 3). Previous studies using non-trait-specific low-density panels also have consistent results [5,24]. This may be because the number of SNPs linked to QTL increased with the increase of SNP number. The linked SNPs have different allele frequencies and different degrees of linkage with QTLs, which may be adverse to genomic prediction [2]. This may be solved by constructing the LD adjusted relationship matrix used in the SNP-heritability model [42], but this is beyond the scope of this study. The three dairy cattle traits used in this study have different genetic architectures. The SCS is controlled by many loci of small effect [43], and the panel with about 30 k SNPs selected by the GenD method gets the best result in genomic prediction (Figure 3A). For panels with 20~50 k SNPs, there is a negative relationship between GEBV accuracy and $\bar{r^{2}}$ or $\sigma_{r^{2}}^{2}$ (Appendix A) in SCS. Therefore, for traits mainly controlled by loci of small effect, considering the genetic distance between SNPs to remain independent between markers may be more advantageous than controlling the physical distance between SNPs. Ye et al. [24] obtained consistent results from the growth and carcass traits in a chicken population. Nevertheless, the superiority of GenD over PhyD in GP is very limited (Figure 3A). For MY and FP, which are affected by loci of moderate or large effect [37], the best genomic prediction results were obtained by using the panels with about 20 k SNPs constructed by PhyD (Figure 3B,C). Previous studies have also shown that the high-density SNP panel was not conducive to the genomic prediction of simulated traits controlled by loci of large effect [25]. For quantitative traits, even if there are loci of large or moderate effect, there may be also a large number of loci of small effect [2]. Therefore, we suggest that the panel with moderate density can be constructed in two steps. First, the PhyD method is used to construct a low-density panel (10 k). Then, based on the low-density panel generated in the first step, selecting the remaining SNPs using the GenD method. For all the traits used in this study, the panels containing 20–30 k SNPs get the best results in GP. Therefore, we suggest using a moderate-density SNP panel (20–30 k) in genomic prediction, and which is close to the marker density commonly used [33]. It is difficult to further improve the accuracy of genome prediction by simply increasing the SNP number on moderate-density panels, but it seems more effective to add trait-related causative variations [4] or construct weighted genome relationship matrices [44,45].

## 5. Conclusions

In this study, the influence of the measurements of ‘marker density’ on the performance of genomic prediction was investigated based on the high-density SNP data of a Holstein dairy cattle population. We found that the smaller the variation degree of physical distance between adjacent SNPs, the more accurate the genomic prediction. There was no significant correlation between the genetic distance between SNPs and the accuracy of genome prediction. However, the variation of physical distance between SNPs selected by genetic distance is large, which is not conducive to genome prediction. The moderate-density panels (20–30 k) constructed by the PhyD method performed in this study are recommended for genome prediction and SNP-heritability estimation. The relationship between measurements of ‘marker density’ on GEBV accuracy discovered in this study provides useful suggestions for future work based on high-density SNP or whole-genome sequence data.

## Figures and Tables

**Figure 1 animals-11-01992-f001:**
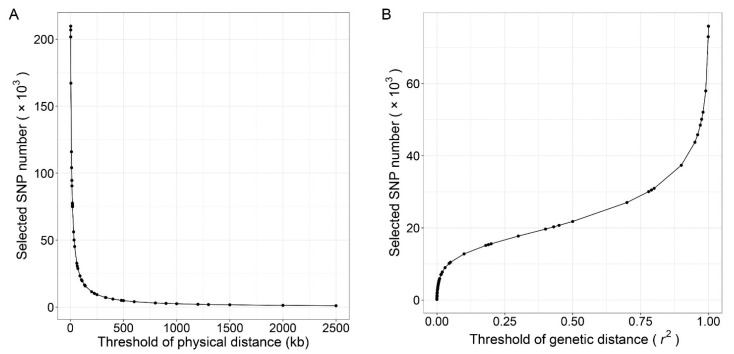
The relationship between the physical distance or genetic distance between adjacent SNPs and the number of SNPs selected. The number of selected SNPs against the threshold of the physical distance between adjacent SNPs is given in left (**A**), and the number of selected SNPs against the threshold of genetic distance (*r*^2^) between each pair of SNPs in the window is given in right (**B**).

**Figure 2 animals-11-01992-f002:**
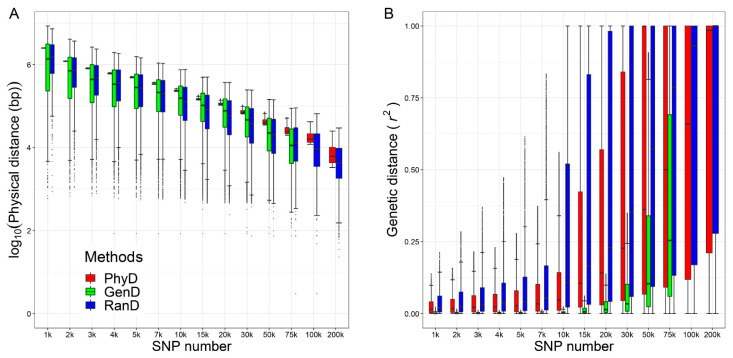
The measurements of ‘marker density’ for panels constructed in this study. The box and whiskers plot of the physical distance between adjacent SNPs (**A**) and the box and whiskers plot of genetic distance between adjacent SNPs (**B**) for panels in different SNP number levels generated by physical distance (PhyD), genetic distance (GenD) and random distance (RanD).

**Figure 3 animals-11-01992-f003:**
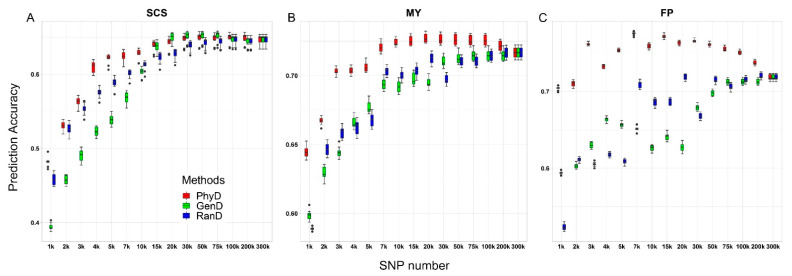
Box plots of genomic prediction accuracy against SNP numbers for three dairy cattle traits. The genomic prediction accuracy of panels with different SNP numbers constructed by PhyD, GenD or RanD for somatic cell score (SCS), milk yield (MY) and milk fat percentage (FP) are shown in (**A**–**C**) respectively.

**Figure 4 animals-11-01992-f004:**
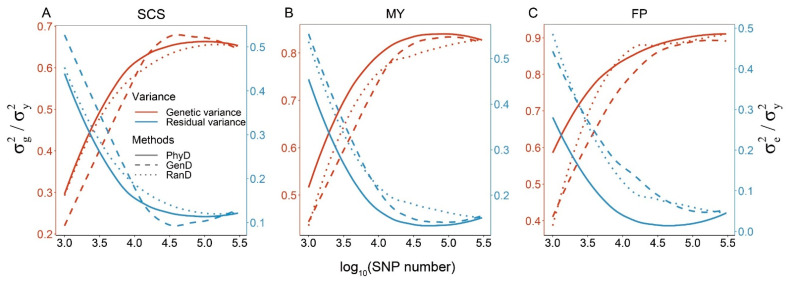
Smoothed conditional means of variance component estimates against SNP numbers for three dairy cattle traits. To facilitate representation, all variance component estimates were scaled with phenotypic variances, i.e., the genetic variance estimates and residual variance estimates were divided by the phenotypic variances, respectively. The smoothed conditional means of variance component estimates of panels with different SNP numbers constructed by PhyD, GenD or RanD for SCS, MY and FP are shown in (**A**–**C**) respectively.

**Figure 5 animals-11-01992-f005:**
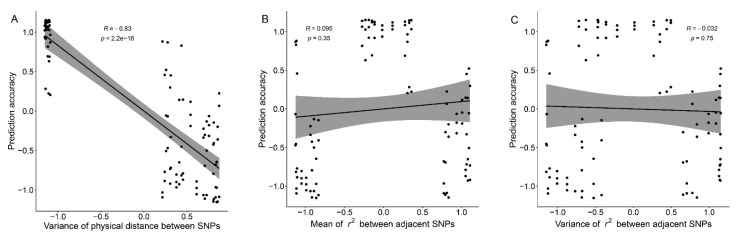
The correlation tests between normalized marker density-related measurements and the normalized genomic prediction accuracy. Correlation tests were performed by combining normalized marker density-related measurements and normalized GEBV accuracies at all SNP number levels. The correlation coefficient, *p*-value of the correlation coefficient generated by each correlation test were given in the plot. The correlation test results of variance of the physical distance between adjacent SNPs, the mean and variance of the *r*^2^ (genetic distance) between adjacent SNPs with genomic prediction accuracy are listed in (**A**–**C**) respectively.

**Figure 6 animals-11-01992-f006:**
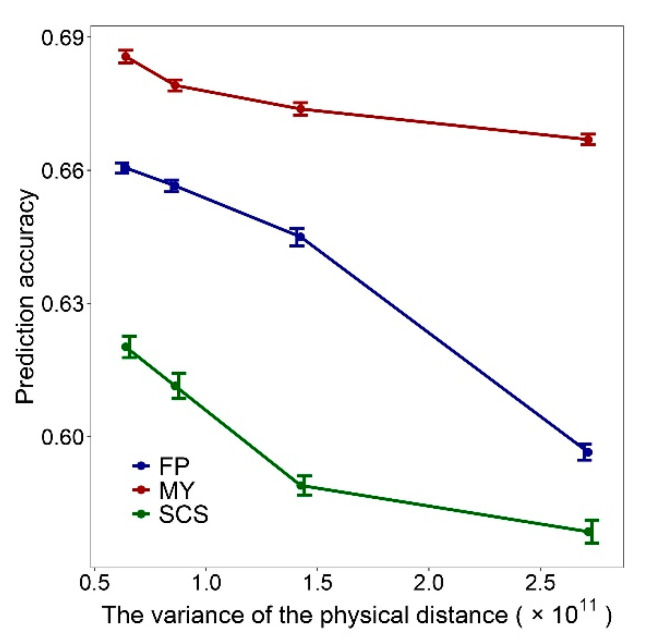
Impact of the variance of physical distance between adjacent SNPs on the genomic prediction accuracy, the standard errors of the genomic prediction accuracy is indicated by the whiskers.

**Table 1 animals-11-01992-t001:** The measurements of marker density and GEBV accuracy for low and moderate-density panels constructed by PhyD, GenD or RanD. Each GEBV accuracy in the table was calculated by a 10 × 10 cross-validation.

SNP Number Levels	Methods	Measurements of Marker Density			GEBV Accuracy		
		$\sigma_{d}^{2}$ (10^10^)	$\bar{r^{2}}$	$\sigma_{r^{2}}^{2}$	FP	MY	SCS
1 k	PhyD	0.245	0.047	0.005	0.704	0.645	0.482
	GenD	867.650	0.027	0.011	0.594	0.599	0.394
	RanD	639.753	0.097	0.038	0.523	0.589	0.459
2 k	PhyD	0.216	0.061	0.012	0.710	0.667	0.531
	GenD	226.149	0.030	0.014	0.603	0.630	0.458
	RanD	158.043	0.125	0.053	0.611	0.647	0.527
3 k	PhyD	0.240	0.082	0.021	0.761	0.703	0.563
	GenD	97.622	0.029	0.014	0.630	0.645	0.491
	RanD	72.558	0.164	0.075	0.605	0.659	0.553
4 k	PhyD	0.268	0.088	0.023	0.732	0.704	0.610
	GenD	52.830	0.030	0.015	0.663	0.666	0.523
	RanD	43.857	0.193	0.091	0.617	0.662	0.576
5 k	PhyD	0.242	0.108	0.034	0.753	0.706	0.621
	GenD	32.881	0.030	0.015	0.656	0.677	0.539
	RanD	28.646	0.219	0.103	0.608	0.668	0.590
7 k	PhyD	0.218	0.138	0.049	0.775	0.720	0.624
	GenD	16.006	0.029	0.014	0.652	0.694	0.569
	RanD	15.380	0.253	0.117	0.709	0.703	0.601
10 k	PhyD	0.212	0.191	0.077	0.759	0.724	0.628
	GenD	6.754	0.029	0.013	0.626	0.692	0.604
	RanD	8.399	0.306	0.138	0.686	0.700	0.612
15 k	PhyD	0.170	0.275	0.114	0.771	0.725	0.640
	GenD	2.976	0.031	0.010	0.641	0.697	0.638
	RanD	4.266	0.373	0.157	0.686	0.703	0.623
20 k	PhyD	0.157	0.323	0.131	0.763	0.727	0.643
	GenD	1.798	0.050	0.009	0.627	0.695	0.650
	RanD	2.627	0.420	0.167	0.718	0.712	0.628
30 k	PhyD	0.128	0.398	0.152	0.765	0.726	0.648
	GenD	0.974	0.110	0.024	0.679	0.710	0.652
	RanD	1.314	0.479	0.174	0.668	0.697	0.640
50 k	PhyD	0.098	0.476	0.167	0.761	0.726	0.650
	GenD	0.485	0.238	0.075	0.698	0.712	0.652
	RanD	0.551	0.553	0.176	0.715	0.710	0.643

## Data Availability

Publicly available datasets were analyzed in this study. The SNP chip data and EBVs of 2000 bulls can be found at: https://www.g3journal.org/content/suppl/2015/02/09/g3.114.016261.DC1 (accessed on 5 February 2015).

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
