# Peer review of "Impact of Marker Pruning Strategies Based on Different Measurements of Marker Distance on Genomic Prediction in Dairy Cattle"

_animals, 2021, doi:10.3390/ani11071992_

Round 1
Reviewer 1 Report
This review considers different strategies for measuring marker density, and using these measurements to select markers on a SNP chip. The accuracy of genomic prediction resulting from these SNP Chips are compared. The article is well written in general, but description of results is sometimes hard to follow. In addition, it seems that a rather ‘complicated’ regression analysis is used that may not be appropriate (but I cannot be sure based on the methods presented). Finally, the paper lacks a clear discussion of the results: are they as expected, or are there surprises? Some major and minor comments are listed below.
Major comments
- The authors use a linear regression model to test for factors (related to marker density) that affect genomic prediction accuracy (lines 212-220). I do not completely understand at what level the regression analysis was performed. The authors mention it is at the SNP number level x trait x measurement situation. I cannot see how many observations are left for each level, and whether observations within a level may be correlated. I therefore wonder whether this regression analysis is even appropriate at all, because the data may be very limited, and model assumptions may be violated (e.g. is the relationship even linear? figure S2 suggests it may not be). Please improve the description of the regression analysis (also include how many observations were available for each model), and explain why you used this approach instead of a more simple approach (e.g. plotting accuracies against measurements, and interpreting those plots).
- The results seem to suggest that accuracy has a positive correlation with mean(d), and a negative correlation with var(d) (Section 4.1). This suggests that the best marker panel is one with very large (and consistent) physical distance between adjacent SNPs. This is surprising to me, especially because there seems to be a positive correlation between marker density and accuracy (at least for low to medium sized panels). Please check whether these conclusions are correct, and if so, please discuss whether you think this was expected.
Minor comments
- Title: The title suggests that genomic prediction accuracy depends on how marker density is measured. However, it does not directly depend on how it is measured, but about how markers are pruned/selected. I would therefore suggest to change the title to something like: “Strategies for marker pruning based on different measurements of marker distance, and their impact on genomic prediction accuracy (maybe add: in dairy cattle).” Please also change your wording throughout the text accordingly.
- which method is used in practice when making SNP chips?
- Abstract: the summary of results (lines 38-50) is difficult to follow. Please improve the structure of this section.
- line 36: please remove “significantly”, it is redundant.
- line 38: Is However the correct word here? This sentence does not seem to contradict the previous sentence.
- Line 41: Can you improve your explanation? accuracy was increased with what compared to what?
- Line 49: Please replace genetic analysis with genomic prediction.
- Line 60: Here (and in other places), sentences start with “And”, which I think should be avoided.
- Line 69: Do you mean However instead of While?
- Line 125-127: This is quite a difficult sentence. Could you split it up to make it more clear?
- Line 131: what do you mean with “generated separately”?
- Line 157: The phenotypic observations are actually EBVs, right?
- Line 179: I would suggest to change the CV strategy slightly. Your current approach may underestimate the uncertainty of accuracies (i.e. the shadows in your line plots).
Instead of computing the accuracy in each fold, you can compute the accuracy across folds: i.e., compute GEBV for animals in the validation set, repeat that for every fold. Then, take the GEBV of all the animals and correlate that with the EBV to get 1 accuracy for that replicate. You can repeat this 10 times to get 10 accuracies for each scenario. - Line 181: Instead of “model”, could you say “scenario”? The model used is actually the same every time.
- Line 187-189: I think you need to give some more information on these statistics. For example, what does r mean (how is it calculated?)
- Line 204-205: I do not understand why joint analysis is discouraged when the magnitudes are different. Could you elaborate?
- Line 227: Something is missing. Larger and smaller than what?
- Line 238: Please introduce the section. What are you going to talk about here?
- Line 240: Figure S2A does not help me to understand the results, especially because you start the section with this. Please move it down or remove it entirely (I do not see why you need it).
- Line 250: It is difficult to see where in the figure is 30k SNPs (because the figure is on the log scale). Please also add the log value.
- Figure 3: First: why did you use a smoothed line? What does the actual curve through means look like? In addition, how can there be observations for GenD at 100k SNPs (I thought you removed those scenarios)?
- Line 257: “cannot appear this inflection point” is unclear to me.
- Line 284-285: This is opposite of expectation, won’t you agree?
- Figure 5: There are descriptions for A and B, not for C and D?
- Figure 6: Panel descriptions are missing.
- Line 323: Something seems to be missing here, please improve.
- Line 324: I cannot follow your argument that mean(d) reflects chromosome coverage.
- Line 370: Similar regression results do not mean that they are the same. Also, I thought you plotted chromosome coverage because you argued it reflects mean(d), and here you say that the plot indicates that mean(d) reflects genome coverage. Seems like circular reasoning.
- Line 409-410: Where do you see a decrease in GEBV accuracy at low density? Compared to what? This does not seem to match with your Figures.
- Line 429: Why do you mention linkage analysis here? That is something different than genomic prediction.
- Line 434: You seem to contradict yourself: before you mentioned that there is no relationship between GEBV accuracy and mean/variance of genetic distance (line 463-464). Can you clarify?
- Line 457: please remove “seems to be more effective”.
Author Response
Thanks for your careful review and constructive suggestions. We have revised this manuscript accordingly. The description of the regression analysis has been extensively revised, and some results have been supplemented and modified

Reviewer 2 Report
This is a good manuscript. The objectives of the research are interesting, the implemented methods are appropriate and well described and the results are clearly shown and well discussed.
I have just a few minor specific comments
Line 17: replace “However” with “Therefore”
Lines 38-39: I do not understand this sentence. Please, reword it
Line 48: replace “but” with “therefore”
Line 60: remove “And”
Line 104: It is not clear enough. I suggest replacing “summary statistics related to ‘marker density’” with “mean and variance of marker distances
You should indicate how many outliers were removed in each case and between adjacent SNPs”
Line 126: Please give the formula for the computation of r2.
Line 181: Remove this sentence. “Finally, the 10 CV accuracies were used to evaluate the performance of the models”. It is indicated in the following sentence with the whole number of tests
maybe the values for the thresholds.
In Figure 1 you show not the variance but the box and whiskers plot of the physical or genetic distance for each SNP number.
Line 257: Replace “cannot appear this inflection point” with “this inflection point does not appear”
Legend figure 4: What do you mean with “conditional means”?
Line 289-291: Sorry, I do not understand what you mean here
Line 319: P=0.3 means that there is not a different from zero relationship not that there is a weak relationship
Line 323-324: this sentence seems incomplete
Line 426-427: “For possible reasons that are explained below” This sentence seems also incomplete
Line 462: remove “and”
Author Response
Thanks for your careful review and constructive suggestions. We have revised this manuscript accordingly.

Reviewer 3 Report
Authors presented results of analysis of "marker density" in cattle population. Performed analysis is very intresting and may bring new insight on use of genomic methods in animal husbandry. Increasing numbers of animals with sequenced genome can provide additional informations and help analyse different method for data mining from genomic database. This may end with improving breeding value and find best way to maximaze it potential. In my opinion this manuscript which is full of many statistical analysis is presented well and clearly. I suggest in line 109 where is German name of institution - Vereinigte Informationssysteme Tierhaltung w.V- add also English name
Author Response
Thank you for your recognition of our research work. For the name of the institution, we write it according to the company's request. we can't find any other English name for this institution. We can only find its full name “Vereinigte Informationssysteme Tierhaltung wirtschaftlicher Verein”

Reviewer 4 Report
Fig. 2 (A) does not contain all the methods' variability assessment at 100k and 200k (see box-and-whiskers plots). Moreover, some genetic distances computed for 10k and higher are outside the scale. Is it suitable for visualization here? Just mention outcomes in the text. Some of the figures (fig. 6, 7) are expected to be scatterplots to test relationship and outliers for each trait separately.
Please give link to Tukey (lines 194-196).
Which algorithms were implemented while calculating PhyD, GenD and RanD?
Author Response

(The authors gave the same response as above.)

Round 2
Reviewer 1 Report
Major comments
The authors have responded elaborately on my major comments, and they have implemented most of my minor comments. My main concern is again with the regression analyses. In my view, the results can be presented more simply, and the regression analysis (especially the second one where you use normalization) makes it way too difficult too follow (and is probably not very reliable considering the number of observations and possibly violated assumptions). I therefore think that the paper would benefit greatly from simply presenting figures of relationships between (normalized) measurements and accuracy, and thoroughly explaining and discussing them. Below is some more explanation and some minor additional comments.
- (related to point 1 in author response). Thank for you explaining the regression analysis in more detail. I understand now that the first regression analysis is at the level of each cell in Table S3. That means that there are only 3 observations, with 10 replicates each (i.e. 10 random cross-validations). I therefore think that a simple linear regression is inappropriate, because the data is very limited, and the analysis does not account for repeated measurements (i.e. correlations between replicates within an ‘observation’). This is especially problematic because you are doing inference on the regression results (for example Figure 5, where you look at coefficients and model fit). In addition, if you only want to know the relationships between measurements and accuracy, and you are not interested in the regression coefficients, then why do regression analysis at all?
Hence, what I would suggest is to create plots such as Figure 7b, where you simply show the relationship between (normalized) marker density measurements and prediction accuracy (do not fit a (smoothed) regression, but simply draw lines and error bars). If your x-axis varies within a group/cell, you could take the mean, or you could even show a simple scatterplot of all observations. In the text, you can describe the relationship you see in the plot (is it positive/negative, does it plateau, etc.). This way, you do not have to do any statistical tests or analysis, but you simply show what the relationship is. - (related to point 2 in author response). There seems to be a misunderstanding with the use of the term ‘coverage’. To me, a high coverage means that the genome is densely covered by a lot of SNPs, so that the distance between SNPs is small. Your explanation seems to suggest that you use the word coverage for ‘the distance a SNP needs to cover before another SNP is found’. I would therefore avoid the term ‘coverage’, especially because it has a different meaning in the context of SNP calling.
You could also stay away from ‘coverage’ all-together: simply explain your results on the relationship between mean(d) and accuracy in the discussion. I do not think that talking about coverage will make it easier for readers to understand. In summary, I would suggest to: (1) plot mean(d) against prediction accuracy (see comment 1), (2) explain what you see in the plot, (3) discuss if the results match your expectation, (4) explain what causes these results: why is there a positive relationship with mean(d) and accuracy?
Minor comments
- Line 55: (related to point 4 in author response) What I meant was, which method is currently used in practice to make SNP chips? In other words, can you explain if one of your methods is related to what is currently done in practice?
- Line 57: SNP panels instead of microarrays?
- Line 133: “limiting the minimum threshold” is redundant. You could say ‘by limiting the number of base pairs between adjacent SNPs’.
- Line 182-188: (related to point 15 in author response) What I meant with my previous comment on CV was to not compute accuracy in step 2, but to do it at the end: 1) randomly split into 10 folds 2) build GP model and estimate GEBV in validation set 3) repeat 2) until each of the folds has served as a validation set, and all animals have a single GEBV 4) compute accuracy as the correlation between GEBV and EBV of all animals (giving 1 accuracy). This process (1-4) can be repeated 10 times, to give 10 accuracies.
- Line 195: Please refer to the Appendix.
- Line 230-231: I did not mention this before, but it seems to be important for your results: it seems quite strange to me that you are looking at the effect of mean(d) on accuracy, but you are reducing differences in mean(d) on purpose because there are some large gaps. You mention that the large gaps affect mean(d) and sigma^2_d, but that’s exactly what you want, right?
- Line 281: (related to point 22 in author response) please also add the log values in other sentences of this paragraph where you mention a SNP density.
- Line 426-427: It is indeed contrary to common sense, but your explanation using coverage is not convincing: I still don’t understand where this result comes from. Could it be due to removing large gaps in computing mean(d)?
- Line 440: Please explain why you think that var(d) may be larger in panels constructed from GWAS results.
- Figure 3: (related to point 23 in author response)Personally, I prefer to see Figure S2 instead of this figure, or use line plots with error bars (as e.g. Figure 7b).
- Table S4: Please indicate if these are accuracies from 1 replicate, or averages of 10 replicates
Author Response
Thank you very much for your valuable suggestions, which make the content of our text clear and more readable. According to your suggestions, we have revised the manuscript in the following aspects:
- Our analysis used the cross-validation method you recommended, but the results were basically the same as the cross-validation results we used before.
- We removed the regression analysis and used a table to show the marker density-related measurements and the GEBV accuracy, and described the relationship in the plots in text. We have also tried to display the contents in Table 1 with line plot, box plot and scatter plot, but none of them could well represent the contents and would cause ambiguity, so we used a table to represent the contents. The correlation between the marker density-related measurements and the GEBV accuracy remains a major concern. Therefore, we tested the correlation between the marker density-related measurement and the GEBV accuracy.
- In this study, the length of the chromosomes used to construct the various panels was fixed. At the same SNP number level, the of each panel is basically the same. In the previous analysis, we removed the large gap of each panel by the IQR method. Due to the different large gaps removed in different panels, the total physical length of different panels, that is, the sum of the physical distances between adjacent SNPs is different. The removal of the large gap makes our results difficult to understand. Meanwhile, removing large gaps is unfair to the comparison of methods, since large gaps may be generated by the marker selection methods. In this revision, we do not remove the large gap. Therefore, of all panels at a certain SNP number level are the same, so we no longer investigate the relationship between and the GEBV accuracy.

Round 3
Reviewer 1 Report
Review on revision 2 of “Impact of marker pruning strategies based on different measurements of marker distance on genomic prediction in dairy cattle”
The paper has improved substantially, but there are still some issues with the interpretation of the results. I have two final comments that I think are important.
- My first comment is on the correlation tests and Figure 5. First of all, I am happy to see you included Figure 5. This clearly shows that in B and C, there is no linear relationship. It therefore does not make sense to do a (linear) correlation test. I think it would be much more valuable to interpret the plot and discuss the observed relationship (and skip the correlation tests). For example, if you look at Figure 5b, there seems to be an optimum. This should be reported and discussed.
Furthermore, I do not understand why you do standardize (not normalize, as you called it!) for each SNP density separately. Please clarify this.
Finally, in Figure 5A, there seem to be two distinct groups, around x=-1.0 and x=0.5. Please make sure that the only difference between these two groups is the variance in physical distance. If these groups, for example, differ in mean(d), than the correlation you see may not be due to difference in the variance of physical distance, but due to differences in mean(d).
- You mention that there is no relationship with genetic distance (for example in line 20 and line 503). I would argue there is indeed a relationship, but it is not linear. This is very obvious in Figure 5 B and C. Please be careful with such statements (check throughout the text).
Author Response
Thank you very much for your constructive comments, which are very important to the readability of our text. We have made the following changes based on your comments.
